# The Role of Glutathione and Its Precursors in Type 2 Diabetes

**DOI:** 10.3390/antiox13020184

**Published:** 2024-02-01

**Authors:** Dawn Tuell, George Ford, Evan Los, William Stone

**Affiliations:** Department of Pediatrics, Quillen College of Medicine, Johnson City, TN 37614, USA; tuell@etsu.edu (D.T.); fordga@etsu.edu (G.F.); losea1@etsu.edu (E.L.)

**Keywords:** glutathione, oxidative stress, type 2 diabetes, reactive oxygen species, N-acetyl-L-cysteine, glycine, prediabetes

## Abstract

Type 2 diabetes (T2D) is a major worldwide health crisis affecting about 6.2% of the world’s population. Alarmingly, about one in five children in the USA have prediabetes. Glutathione (GSH) and its precursors play a promising role in the prevention and management of type T2D. Oxidative stress (OxS) is a probable factor in both T2D initiation and progression. GSH is the major cytosolic water-soluble chemical antioxidant and emerging evidence supports its role in improving T2D outcomes. Dietary supplementation with N-acetyl-cysteine (NAC) and/or glycine (GLY), which are GSH precursors, has also been studied for possible beneficial effects on T2D. This review will focus on the underlying pathophysiological and molecular mechanisms linking GSH and its precursors with T2D and OxS. In addition to their traditional antioxidant roles, the in vivo effects of GSH/NAC/GLY supplements will be evaluated for their potential abilities to modulate the complex pro-oxidant pathophysiological factors (e.g., hyperglycemia) driving T2D progression. Positive feedback loops that amplify OxS over long time intervals are likely to result in irreversible T2D micro- and macro-vascular damage. Most clinical studies with GSH/NAC/GLY have focused on adults or the elderly. Future research with pediatric populations should be a high priority since early intervention is critical.

## 1. Introduction

The objective of this review is to critically evaluate the role of glutathione (GSH) and its chemical precursors (N-acetyl-cysteine (NAC) and glycine (GLY)) in the prevention and management of T2D and prediabetes. Oxidative stress (OxS) is a probable factor in both T2D initiation and progression [1]. Antioxidant-based therapy may, therefore, have potential benefits [2]. An ever-increasing body of evidence supports the role of OxS in T2D pathogenesis [3,4,5]. OxS is usually defined as Sa physiologically significant alteration in redox status resulting from an overproduction of reactive oxygen species (ROS) and/or a deficiency in antioxidant protective mechanisms [6]. It is important, however, to also consider how antioxidants, such as GSH, interconnect with the complex pathophysiological risk factors that modulate T2D redox status [7]. GSH is the major cytosolic water-soluble chemical antioxidant and clinical evidence suggests that patients with T2D have a GSH deficiency [8]. As detailed below, GSH is important in the glutathione peroxidase (GPX) system where it reduces hydroperoxides, which are prooxidants very relevant to T2D initiation and progression [9].

### 1.1. The Epidemiology of T2D

The alarming present and forecasted global prevalence of T2D has stressed the need to evaluate novel treatment strategies that could complement lifestyle modifications and conventional pharmacological treatments [10,11,12]. It has been projected that by 2050, some 1.3 billion people worldwide will have diabetes with about 96% having T2D [13]. Both T2D and prediabetes are rapidly increasing in pediatric populations [14,15]. This trend is particularly worrisome since neither the natural history of T2D progression nor the efficacy of pharmacological treatments are as well characterized in pediatric populations compared to adults [16]. In prediabetes, blood glucose levels have not yet reached the T2D threshold [17]. In the United States (US), it has been estimated that more than one in three adults and more than one in five children have prediabetes [15,17]. 

The US Preventive Services Task Force (USPSTF) recently affirmed screening for prediabetes or T2D in adults with overweight or obesity but did not extend this recommendation to children and adolescents [18]. The USPSTF concluded that “the evidence is insufficient to assess the balance of benefits and harms of screening for T2D in children and adolescent” [19]. In contrast, a recent volunteer Task Force of medical experts has suggested screening for prediabetes and diabetes in children, adolescents, and adults who have obesity [20]. The American Academy of Pediatrics and the American Diabetes Association have both recently recommended screening for abnormalities in glucose metabolism in children with a body mass index (BMI) at or above the 95th percentile [21,22]. A key rationale for screening children and young adults is that early and intensive intervention will reduce future morbidity and mortality [20]. Obesity has long been recognized as a major risk factor for cardiovascular disease, prediabetes, and T2D, as well as promoting OxS and markers of inflammation [23,24,25]. 

### 1.2. The Etiology of T2D

T2D is a chronic disease in which blood glucose levels are high enough and of sufficient duration to cause damage to susceptible organs/tissues. Insulin resistance, pancreatic beta-cell dysfunction, and abnormally high glucagon levels are the major etiological factors that cause high blood glucose levels [26,27]. The progression of T2D starts with insulin resistance followed by prediabetes, overt T2D with fasting blood glucose level above 126 mg/dL, and finally vascular damage [7,27]. T2D is, however, a multifactorial disease with alterations in lipid and protein metabolism as well as complex interactions between organs, tissues, cell types, and subcellular compartments/organelles. In this regard, it is significant that alterations in lipid metabolism (e.g., high triglyceride levels) have been found to occur years before overt T2D diagnosis in adults and can improve the prediction of T2D progression [28]. Gummesson et al. [29] have observed distinct alterations in plasma protein profiles in newly diagnosed adult T2D patients in comparison to healthy adult controls. These data suggest that early-stage T2D is accompanied by plasma protein alterations that may also prove useful in helping to predict T2D progression. 

The complexity of T2D is reflected by the fact that it is a polygenic disorder with an association with over 120 genetic loci [30]. In addition to insulin resistance, obesity (and intraorgan adipose tissue), poor-quality high-calorie diets, and lack of physical exercise are major risk factors for T2D [26]. The consumption of high-calorie/high-fat ultra-processed foods by children and adolescents is strongly associated with obesity [31,32] and obesity, in turn, is associated with chronic OxS [25]. OxS has emerged as an important mechanism for the initiation of prediabetes and for promoting T2D progression [3,4,7]. In Section 2 of this review, we outline the biochemistry of GSH and its central role as an antioxidant important in modulating OxS and T2D. With this background in place, we mechanistically link (in Section 3) T2D risk factors to OxS and T2D initiation/progression. 

## 2. The Biochemistry and Roles of GSH (and Its Precursors) in OxS and T2D

GSH is a key water-soluble thiol (R-SH) antioxidant that may play an important role in preventing, slowing, and perhaps reversing T2D progression [33]. In 1969, Kosower and Kosower wrote a review of GSH with the title “Least I Forget Thee, Glutathione” [34]. This admonition has been well heeded as evidenced by an eight-fold increase in the number of GSH publications in 2022 compared to 1969 [35]. GSH (gamma-glutamylcysteinylglycine) is most often described as a “tripeptide” composed of CYS, GLY, and glutamic acid (GLU) as shown in Figure 1. Both GLY and CYS are considered conditionally essential amino acids in the context of T2D [36,37].

While it is true that GSH is a tripeptide, it is not a “eu-tripeptide”. Eupeptides have amide bonds that are formed between the C-l of one amino acid and the N-2 of another amino acid as occurs in typical proteins. In GSH, there is an iso-peptide bond between the gamma-carboxyl group of the GLU side chain and CYS (see Figure 1). This iso-peptide bond is important because it renders GSH relatively resistant to intracellular proteases that cleave eu-peptide bonds [38]. This intracellular protease resistance enables GSH to reach unusually high intracellular concentrations (5 mM) for an organic compound and thereby contributes to its effectiveness as an intracellular antioxidant [39]. In contrast, the plasma level of GSH in healthy individuals is about 0.0034 mM [40]. 

### 2.1. Diminished GSH Synthesis in T2D Results in OxS

Patients with uncontrolled T2D (i.e., persistent hyperglycemia) have been found by Sekhar et al. to have severely diminished in vivo GSH synthesis, which can be restored by dietary supplementation with “GSH plus GLY” [41]. It should be noted that the dietary “GSH” used in this research was provided as NAC, which is a xenobiotic precursor to GSH; it would be more accurate, therefore, to state that “NAC plus GLY” was used as the dietary supplement. Surprisingly, the mechanism by which NAC enters cells is not completely understood but NAC cannot simply be considered the equivalent of GSH [35]. Significantly, Sekhar et al. found that supplementation with “NAC plus GLY” reduced measures of systemic OxS, e.g., serum lipid peroxides (see Section 2.2.1 below) [41]. 

GSH is present and synthesized in the cytosol of all mammalian cells by a two-step regulated process (see Figure 1) [42]. In the first step, L-glutamate-L-cysteine ligase (GCL) catalyzes the formation of gamma-L-glutamyl-L-cysteine (GGC) by linking CYS (a thiol-containing amino acid) and GLU. In the second step, glutathione synthetase (GS) catalyzes the formation of GSH by linking GGC to GLY [38,43]. The first step is rate-limiting for GSH synthesis, and the availability of CYS and the activity of GCL are key determinants of GSH synthesis [42]. CYS, and its thiol group, are essential for the antioxidant properties of GSH [44]. Moreover, since the Km of CYS for the catalytic subunit of GCL is close to its cytosolic concentration, a drop in CYS levels would result in decreased GSH synthesis [44]. Intracellular levels of CYS are typically low and, if the demand for GSH is high (e.g., OxS), CYS will be transported into the cell from the extracellular space [45]. There are sound reasons for utilizing NAC rather than CYS as a dietary supplement since NAC is chemically more stable than CYS and has an excellent safety record [35]. The enzymes for GSH synthesis are not present in mitochondria but two transport systems pump GSH from the cytosol into these organelles [46]. GSH is thought to act both as a direct antioxidant by quenching reactive oxygen species (ROS) and as a cofactor for GPX [47]. 

### 2.2. GSH and the Glutathione Peroxidase (GPX) System

As outlined in Figure 2, GPX enzymes reduce hydroperoxides utilizing hydrogen ions donated by the thiol group of GSH with the formation of oxidized GSH (GSSG). There are eight known GPXs (GPX1-GPX8) with GPX1 being the most abundant and this isoform is expressed in the cytoplasm and mitochondria of most cells [48,49]. GPX1-4 and GPX6 are selenoenzymes with selenocysteine (selenium replacing sulfur in CYS) at the active site. Selenium (Se) is an essential trace element and in the absence of dietary Se, total GPX activity in most mammalian tissues is markedly decreased [50]. GPX1-4 and GPX6 can reduce either organic hydroperoxides (ROOH) or hydrogen peroxide (H_2_O_2_) to the corresponding organic alcohol or H_2_O in the case of H_2_O_2_. 

GPX1, due to its affinity for H_2_O_2_ and its abundance in most cells, is the key enzyme responsible for minimizing cytosolic H_2_O_2_ levels as well as mitochondrial H_2_O_2_ levels where no catalase is present in most mammalian cells [49,51,52]. Mitochondria, in addition to GPX1, also contain GPX4, which is unique in its ability to reduce phospholipid hydroperoxides (PLOOH), which are not water-soluble [49]. As indicated in Figure 2, oxidized GSH is recycled back to GSH by glutathione reductase (GR) with the consumption of nicotinamide adenine dinucleotide phosphate (NADPH). An overproduction of H_2_O_2_ or lipid hydroperoxides can result in a reduced GSH/GSSG, ratio which is often used as a biomarker for OxS [53]. Under normal physiological conditions, about 98% of the total cellular GSH (GSH+GSSG) content is GSH. 

While Se is required for most forms of GSPX, this trace element is present in some 25 other selenoproteins [54]. The role of Se in T2D has been studied for many decades yet remains controversial [55]. A study by Laralis using a small population of patients (N = 94) with T2D (no diabetic complications and consuming a Mediterranean diet) suggests that 200 micrograms/day (chemical form not specified) can improve glycemic control (after three or six months) [56]. Nevertheless, there is now a consensus suggesting that the effect of Se intake on hyperglycemia has a U-shaped dose–response curve, i.e., a dose above the recommended level (55 micrograms/day) causes hyperglycemia (and hyperinsulinemia) [55,57]. Although beyond the scope of this review, Se has been found to modulate the insulin signaling pathway (see below) [55]. 

Blood levels of GPX are lower in patients with T2D compared to healthy controls and the decrement in GPX activity was more severe in T2D patients with obesity compared to T2D patients without obesity [58]. Compared to healthy controls, patients with T2D have lower levels of red blood cell GSH levels as well as lower GSH synthesis rates, particularly in T2D patients with microvascular complications [8]. 

#### 2.2.1. Lipid Peroxidation, Protein Carbonylation, and T2D

Lipid hydroperoxides (LOOH) formed by the process of lipid peroxidation are also a biomarker of OxS. If not reduced by the GPX system, lipid hydroperoxides can accumulate and decompose, yielding lipid peroxyl radicals (LOO*) and/or lipid alkoxyl radicals (LO*), which can amplify lipid peroxidation with the formation of additional lipid hydroperoxides [59]. The positive associations between lipid peroxidation and T2D progression have been comprehensively reviewed by Shabalala et al. [9]. Both malondialdehyde (MDA) and 4-hydroxy-2-nonenal (4-HNE) are reactive aldehyde–lipid peroxidation by-products (see Figure 2) that have been widely used as biomarkers of OxS [59,60]. A meta-analysis by Bank and Ghosh found that serum levels of MDA were higher in subjects with T2D compared to controls [61]. Similarly, increased serum (as well as other biofluids and tissues) levels of 4-HNE have been documented in T2D patients compared to controls [60]. Plasma levels of MDA and 4-HNE are associated with T2D progression and poor glycemic control [9]. 

As indicated in Figure 2, MDA and 4-HNE can covalently modify proteins by carbonylation of CYS, histidine (HIS), and lysine (LYS) residues, thereby potentially altering their structure and function(s), e.g., signal transduction pathways [62,63]. CYS, HIS, and LYS are often present at the active sites of many enzymes. Human ex vivo studies as well as studies in animal models suggest that obesity and insulin resistance are linked to increased levels of adipose protein carbonylation [64,65]. In an animal model of obesity and insulin resistance, it has been estimated that about 6–8% of adipose proteins are modified by carbonylation, i.e., a nontrivial level [60,64]. The potential for protein carbonylation to inactivate key antioxidant enzymes has not been well studied despite the potential relevance to T2D etiology. Interestingly, carbonylated GPX1 and peroxiredoxin1 (PRX1) have both been identified in adipose tissue but the functional significance of this modification is not known [64]. PRXs are a family of peroxidases (like GPXs) that can reduce H_2_O_2_ or ROOH and are thought to be important in protecting beta-cells from oxidative damage and diminished insulin secretion [66,67]. Remarkable analytical progress in detecting and characterizing carbonylated proteins, particularly in the area of proteomics, suggests that future studies will be forthcoming [4,60]. 

### 2.3. NAC Metabolism and Its Role as an “Antioxidant” in T2D Management

NAC has been described as the “most frequently used” antioxidant supplement but recent research suggests that its precise mechanism of action is less certain [35,68,69]. NAC supplements are traditionally thought to provide the CYS residues needed to support GSH synthesis [35]. It is likely that NAC is not an effective direct antioxidant and must be converted into GSH and/or hydrogen sulfide and sulfane sulfur species [35,70]. NAC is hydrolyzed to CYS by aminoacylase1 (ACY1) (Figure 1), which is found in many tissues including the liver and intestines [71]. CYS, in turn, promotes GSH synthesis primarily under circumstances in which tissue GSH is depleted, e.g., T2D [35]. The newly synthesized GSH can act as an antioxidant via the GPX system (Figure 2) [72]. 

Pedre et al. [35] have reviewed the evidence suggesting that NAC-derived CYS can also exert antioxidant activity by conversion into hydrogen sulfide (H_2_S) and sulfane sulfur species, which may act as antioxidants. The conversion of CYS to H_2_S depends on cystathionine-gamma-lyase (CSE), which is normally present only in the cytoplasm but under stress simulation is translocated to the mitochondria and supports H_2_S production in these organelles [73]. It is significant (see below) that the oxidation of H_2_S into sulfane sulfur species occurs primarily within mitochondria since OxS in these organelles has been linked to transitory insulin resistance [74]. 

Szkudlinska et al. [75] tested the hypothesis that short-term (4 weeks total) NAC supplementation (no GLY) might improve glucose tolerance and/or beta-cell function in T2D. The study population (N = 12) in this report was small and patients with severe hyperglycemia were excluded. Nevertheless, no benefit in glycemic control, glucose tolerance, insulin resistance, or oxidative stress markers was observed [75]. In contrast, a small pilot study by Sekhar et al. found that an “NAC plus GLY” supplement for two weeks lowered insulin resistance in T2D subjects (N = 10) [33]. This author has made a compelling argument that “NAC plus GLY” is uniquely different from “NAC alone” or “GSH-alone” in so far as GLY is necessary for GSH synthesis [33,76]. An experiment in which NAC alone, GLY alone, GSH alone, NAC plus GLY, and GSH plus GLY would be needed to affirm this assertion. In older normal adults (N = 24) with no known T2D, Kumar et al. found that NAC plus GLY (for a total of 16 weeks) improved GSH deficiency, reduced markers of OxS, and improved insulin resistance [76]. 

### 2.4. GLY Alone Has Been Found to Play a Role in Promoting Insulin Resistance 

It has long been noted that GLY deficiency (hypoglycinemia) is associated with obesity or T2D and that improvement in insulin resistance is associated with increased plasma GLY [77]. In a comprehensive literature review, McCarty et al. [78] concluded that dietary GLY is rate-limiting for GSH synthesis and that supplemental GLY might promote GSH synthesis and be clinically effective (and safe) in health disorders in which OxS is relevant, e.g., T2D. For over a decade, it has been known that dietary collagen supplementation (rich in GLY) strongly potentiates glucose-stimulated insulin secretion in patients with T2D [79]. In an animal model of sucrose-induced insulin resistance, it has been demonstrated that dietary GLY supplementation decreases liver OxS biomarkers, increases liver GSH, and improves insulin sensitivity [80]. Since GLY has no intrinsic antioxidant activity, it is likely that that its in vivo beneficial effect on OxS and insulin resistance is indirect and mediated, at least in part, by increased GSH synthesis. 

A second mechanism by which GLY could affect glucose homeostasis lies in its potential ability to modulate insulin secretion from pancreatic beta-cells by activating ligand-gated chloride channels [78,81]. A progressive decline in insulin secretion from beta-cells is a hallmark of T2D progression [27]. A small clinical study (N = 9) in 2002 found that GLY supplementation increased plasma insulin levels in healthy subjects but did not establish a mechanism [79]. In pioneering work, Yan-Do et al. [82] found that GLY binding to GLY receptors (GLYRs) on beta-cells stimulates insulin secretion by promoting an inward Ca^2+^ flux. Moreover, these authors found that GLYR expression and GLY-induced currents are reduced in beta-cells from T2D donors, thereby contributing to impaired insulin secretion [82].

## 3. Interconnections between OxS, T2D Risk Factors, and GSH Metabolism

Insulin resistance and impaired beta-cell secretion of insulin both contribute to hyperglycemia-induced OxS and diabetic complications [83]. A paper by Boyaci et al. [84] in 2021 raised the question of whether OxS is a consequence of hyperglycemia or if hyperglycemia is a result of OxS. It is likely, however, that multiple “positive feedback loops” between T2D risk factors and OxS are at play and drive T2D progression in susceptible individuals. A general review of positive feedback loops in biological systems has been published by Mitrophanov et al. [85]. 

Over long time intervals, T2D positive feedback loops can eventually result in irreversible OxS micro- and/or macro-vascular damage such as diabetic retinopathy (a microvascular disease) [86]. Figure 3 provides a simplified scheme showing the interconnections between OxS and T2D risk factors as well as some likely positive feedback loops. High-calorie meals, overweight/obesity, gastrointestinal postprandial oxidative stress (POS), insulin resistance, hyperglycemia, and pancreatic beta-cell dysfunction are examples of risk factors that can modulate both local and systemic OxS in T2D. Encouragingly, accumulating evidence suggests that T2D progression is not “inevitable” and possibly reversible at an early stage [26]. As will be detailed below, GSH metabolism plays a central role in modulating T2D OxS and may play a role in reversing/slowing early T2D progression. 

In addition to high-calorie meals, it should be mentioned that there are other common exogenous sources of OxS. As comprehensively reviewed by Bhattacharyya et al. [87], some pollutants, radiation, cigarette smoking, and some drugs/xenobiotics can contribute to OxS. As recently noted by the World Health Organization, quitting cigarette smoking can decrease the risk of developing T2D by 30–40% [88]. In addition to mitochondrial OxS (as detailed below), there are numerous sources of endogenous OxS (e.g., inflammatory responses) that can be relevant to T2D progression [87]. 

### 3.1. High-Fat/High-Calorie Diets Promote Postprandial Oxidative Stress (POS), Mitochondrial OxS, and Insulin Resistance

In healthy young adults, the consumption of a lipid (saturated fat)-rich meal results in robust postprandial oxidative stress (POS) compared to an isocaloric carbohydrate-rich meal (dextrose) [89]. When fed a standard meal, patients with T2D show a greater POS compared to matched healthy subjects [90]. POS has been proposed as a probable mechanism contributing to systemic OxS, T2D progression, and vascular damage (see Figure 3) [91,92]. Moreover, supplementation (for 15 days) with dietary antioxidants (including NAC) has been found to reduce POS and improve markers of endothelial dysfunction in subjects with T2D or insulin resistance [93]. In addition to promoting OxS, high-calorie/fat diets are also problematic because they are likely an initiating factor for prediabetes (see below) and promote overweight and obesity. In marked contrast, very low-calorie (hypocaloric) diets can improve glycemic control in T2D patients and potentially promote T2D remission [94].

### 3.2. Skeletal Muscle Mitochondrial OxS and Insulin Resistance

It has long been recognized that skeletal muscle insulin resistance is the primary defect in T2D [95]. The insulin-sensitive glucose transporter 4 (GLUT4) is the primary means by which skeletal muscle takes up glucose. Insulin stimulates the translocation of GLUT4 from intracellular GLUT4-containing vesicles to the cell surface where this transporter can actively support facilitated glucose transport [96]. The insulin signaling pathway is complex but requires the activation of Akt (a serine–threonine kinase) by phosphorylation [97]. The reduced ability of insulin to activate the GLUT4 glucose transport system in skeletal muscle is a primary cause of insulin resistance [96,98]. 

In pioneering work, Anderson et al. [74] found that a high-calorie diet (in healthy adults) promoted skeletal muscle mitochondrial OxS which, in turn, resulted in transient insulin resistance. While both a high-calorie fat or carbohydrate meal could induce this transient insulin resistance, dietary fat was more effective than dietary carbohydrates [74]. Obesity, a known risk factor for T2D, dramatically amplified skeletal muscle mitochondrial OxS [74]. As indicated in Figure 3, positive feedback loops connecting high-calorie/fat meals, overweight/obesity, mitochondrial OxS, skeletal muscle insulin resistance, increased glycemia, and systemic OxS are likely at play in T2D. In susceptible individuals, it is likely that these positive feedback loops promote T2D progression. In addition to the importance of lifestyle and environmental factors, T2D has an inheritance ranging from 30 to 70% [99]. 

The work by Anderson et al. [74] is also important since these investigators demonstrated that a high-fat diet induced skeletal muscle OxS (and insulin resistance) by promoting mitochondrial H_2_O_2_ emission. Although not the primary focus of their work, Anderson et al. [74] found that glucose-stimulated Akt phosphorylation in skeletal muscle was markedly diminished by a high-fat diet, suggesting a disruption in the insulin signaling pathway. The high-fat diet also induced a decrease in the skeletal muscle GSH/GSSG ratio, suggesting that GSH was oxidized to GSSG by GPX in response to increased H_2_O_2_ production. It should again be noted that mitochondria do not have the enzymes required for GSH synthesis and must import GSH from the cytoplasm [46]. GPX enzymes were not studied by Anderson et al. but GPX1 and GPX4 are pivotal since they reduce mitochondrial H_2_O_2_ [49]. Recent research indicates that genetic variants in GPX1 and GPX3 are associated with T2D risk [100]. 

Søndergård et al. [101] conducted a small clinical trial looking at the effects of three weeks of oral GSH supplementation on whole-body insulin sensitivity in obese subjects with (N = 10) and without (N = 10) T2D. These investigators [101] found that GSH supplementation increased insulin sensitivity in both the obese subjects with and without T2D but did not change the GSH/GSSG ratio in skeletal muscle as might have been anticipated by the work of Anderson et al. [74]. As mentioned above (Section 2.3 and Section 2.4), simultaneous GLY supplementation could be important in promoting an optimal redox status. 

### 3.3. Hyperglycemia Promotes Protein Glycation, Formation of AGEs, Activation of the Polyol Pathway, OxS, and T2D Progression

Chronic bouts of transient insulin resistance will result in chronic increases in postprandial glucose (PPG) as indicated in Figure 3. Glucose can covalently react with lysine, arginine, and the N-terminal residues on proteins to form glycation products, which can further react to produce advanced glycation end products (AGEs) [102]. Glycation can modify the structure and functions of proteins. It has been found, for example, that glycation of the GPX results in a loss of enzymatic activity [103]. This is a hypothetical example of a positive feedback loop, i.e., increased mitochondrial OxS promotes increased glycemia, promoting increased glycation–inactivation of GPX enzymes with a further increase in skeletal muscle mitochondrial OxS and glycemia. 

As reviewed by Sottero et al. [92], increased plasma levels of glucose resulting from insulin resistance can promote the formation of AGEs. Hyperglycemia will eventually result when beta-cell secretion of insulin is unable to make up for insulin resistance [104]. AGEs can stimulate ROS production as a result of binding to RAGE (receptor for AGEs) [105]. In vitro and ex vivo evidence (animal models and T2D patients) supports the role of the AGE-RAGE pathway in promoting OxS-driven beta-cell dysfunction and decreased insulin secretion [106,107]. Evidence from an animal model also shows that AGEs can induce insulin resistance by repressing the skeletal muscle glucose transporter GLUT4 [108]. Repression of GLUT4 is known to contribute to insulin resistance [95,98]. As indicated in Figure 3, these sequences of molecular events are also another potential set of positive feedback loops. 

High plasma glucose also promotes OxS by activation of the polyol pathway in which aldose reductase (AR) reduces glucose to sorbitol followed by the conversion of sorbitol to fructose by sorbitol dehydrogenase (SDH). In the first reaction, NADPH is consumed by AR, which can result in OxS since NADPH is required for reducing GSSG to GSH (by GR) and GSH is required for the reducing peroxides by the GPX system (see Figure 2). It has been estimated that in diabetes, as much as 30% of body glucose can be consumed by the polyol pathway [102]. 

For cells primarily relying on insulin-independent GLUT transporters, such as GLUT1, plasma levels of glucose will equilibrate with intracellular cytosolic glucose. When postprandial glucose and/or fasting blood glucose are sufficiently high, the polyol pathway (and OxS) will be activated in cells relying on GLUT1, e.g., endothelial cells [109,110]. Moreover, the fructose produced by the second reaction in the polyol pathway is a much more effective glycating agent than glucose [102,111]. Dietary fructose has been implicated in promoting T2D [111]. Both microvascular and macrovascular endothelial cells express GLUT1 and early endothelial dysfunction in T2D is thought to be a driver of future cardiovascular disease [112]. The role of hyperglycemia-induced OxS damage to endothelial cells has been reviewed [110]. The activation of the polyol pathway and the resulting OxS have been strongly implicated in the development of diabetic retinopathy [113].

### 3.4. Chronic Inflammation, OxS, and T2D

OxS arising from chronic inflammation has long been recognized as a potential driver of T2D progression [114,115]. Chronic inflammation is associated with increased levels of ROS, reactive nitrogen oxide species (RNOS), and C-reactive protein. As reviewed by Son et al. [116], the increased production of superoxide radicals (O_2_^•−^) resulting from T2D hyperglycemia can rapidly react with nitric oxide (NO) to produce peroxynitrite (ONOO^−^), which can subsequently react with protein (and apolipoprotein) tyrosine residues to form 3-nitro-tyrosine (3-NT). 3-NT levels are biomarkers for inflammation and patients with T2D have increased serum 3-NT levels compared to controls [117]. ROS and RNOS are both thought to contribute to T2D macro- and micro-vascular damage [116]. GSH is thought to play a key role in NO biochemistry [118]. GSH, for example, can react with NO to form S-nitrosoglutathione, which may be relevant to T2D by promoting insulin sensitivity [119]. 

Serum levels of C-reactive protein are clinically used as a measure of systemic inflammation and are increased in T2D [120,121]. Although not focused on T2D, a meta-analysis by Askari et al. [122] found that oral NAC supplementation reduced serum levels of C-reactive protein. In a small study of T2D subjects (N = 24 adults), Jeremias et al. [123] found that oral NAC supplementation (four weeks) significantly reduced C-reactive protein levels compared to placebo. As suggested by these researchers, “further study” is well justified. We would also suggest the inclusion of a pediatric population. 

## 4. Is NAC a Drug and/or a Dietary Supplement?

Despite the potential clinical benefits of NAC+GLY for slowing and perhaps even reversing T2D progression, there is a practical issue in its clinical utilization, i.e., it is not entirely clear if NAC is a drug or dietary supplement from a regulatory point of view. The US Federal Food and Drug Administration (FDA) has asserted that NAC was approved as a drug (in 1963) before it was promoted as a dietary supplement and therefore cannot be considered a dietary supplement. According to the National Institutes of Health (NIH), “supplements are products intended to supplement the diet. They are not medicines and are not intended to treat, diagnose, mitigate, prevent, or cure diseases” [124]. In addition, “Medicines must be approved by FDA before they can be sold or marketed. Supplements do not require this approval”. Nevertheless, in 2022, the FDA decided to exercise “enforcement discretion” over NAC and this product is freely available (as of October 2023) for purchase without a prescription [125]. It has been emphasized that stronger federal oversight and regulation of dietary supplements is desirable and could result in enhanced public/physician confidence concerning safety and research outcomes [126,127].

## 5. Conclusions

OxS plays a central role in the initiation and progression of T2D and dysfunction of the GPX system due to low GSH levels is a key mechanism giving rise to systemic and tissue-specific OxS. Deficient GSH synthesis resulting from decreased levels of GSH precursors is a key driver of T2D initiation and progression. Several clinical studies, albeit with relatively small study populations, have shown that GSH, and its precursors NAC and GLY, can reduce OxS biomarkers and lower insulin resistance. These studies have almost exclusively been conducted with adult populations in which the disease burden is already present. Positive feedback loops that amplify OxS over long intervals can eventually result in irreversible OxS-driven micro- and macro-vascular damage. These positive feedback loops reinforce the notion that early intervention is the optimal strategy for potentially reversing or dramatically slowing T2D progression. NAC and GLY have excellent safety records and long-term double-blind placebo-controlled studies with a pediatric population should be a high future priority. 

## Figures and Tables

**Figure 1 antioxidants-13-00184-f001:**
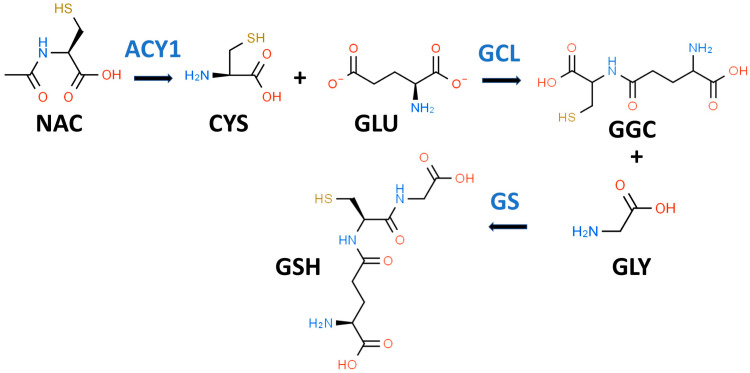
Glutathione biosynthetic scheme. N-acetyl-L-cysteine (NAC) can supply cysteine (CYS) for the biosynthesis of reduced glutathione (GSH). NAC must first be hydrolyzed by aminoacylase 1 (ACY1) to release CYS. In the first step of GSH synthesis, L-glutamate-L-cysteine ligase (GCL) catalyzes the formation of gamma-L-glutamyl-L-cysteine (GGC) by linking CYS and L-glutamate (GLU). In the second step, glutathione synthetase (GS) catalyzes the formation of GSH by linking GGC to GLY.

**Figure 2 antioxidants-13-00184-f002:**
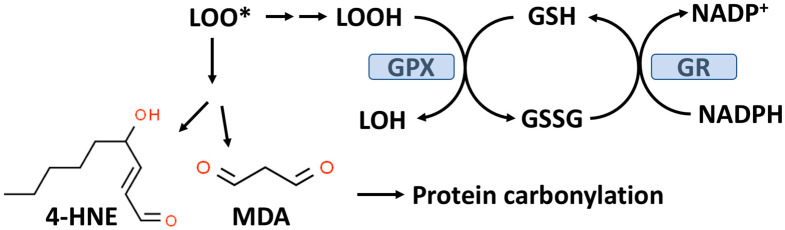
The glutathione peroxidase (GPX) system and lipid peroxidation. GPX catalyzes the conversion of lipid hydroperoxides (LOOH), formed from lipid peroxidation to a lipid alcohol (LOH) utilizing GSH as a reducing agent. The oxidized GSH (GSSG) formed by this reaction is reduced back to GSH by glutathione reductase (GR) with the consumption of NADPH. The lipid peroxyl radical (LOO*) formed from lipid peroxidation can undergo chemical decomposition to 4-hydroxynonenal (4-HNE) and malondialdehyde (MDA), which are reactive aldehydes that can react with proteins to form carbonylation products.

**Figure 3 antioxidants-13-00184-f003:**
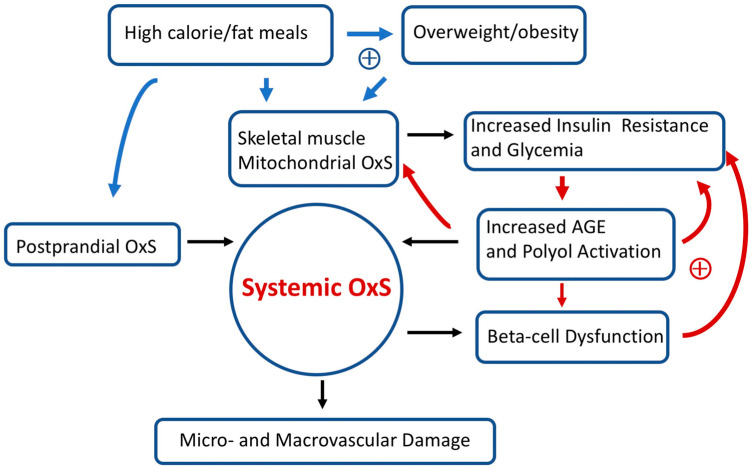
Simplified scheme connecting oxidative stress (OxS) and T2D risk factors. Two sets of potential positive feedback loops (circles with +) are indicated (blue and red arrows). The text describes these positive feedback loops in more detail. Over prolonged time intervals, systemic OxS can cause beta-cell dysfunction as well as irreversible micro- and macro-vascular damage.

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
