# Peer review of "The Role of Glutathione and Its Precursors in Type 2 Diabetes"

_antioxidants, 2024, doi:10.3390/antiox13020184_

Round 1

Reviewer 1 Report

Comments and Suggestions for Authors

The manuscript  brings The Role of Glutathione and its Precursors in Type 2 Diabetes a new and an important inside in diabetes mellitus type 2 pathogenesis and treatment.

Observations:

Introduction

-        Line 66 – before you mention protein and lipids alteration, please give details about carbohydrate metabolism in DM and why the the alteration of protein and lipids metabolism is onset and when.

-        Line 203 – please abbreviate NAC before using it. In the title I suggest do not use abbreviations.

-        Line 234 – same observation NAC

-        Line 258 – I would mention hyperglycemia induces OxS and diabetic complications, also as a reveres process (OxS induces hyperglycemia). In this manuscript  you describe how to reverse hyperglycemia by antioxidant molecules, therefore I consider is more important to mention and describe the importance of OxS as mechanism of onset and progression of diabetes mellitus. Indeed, in the text below you describe multiple loops between these processes, but I would emphasize more OxS indeced hyperglycemia as a main mechanism, because this is the subject of your manuscript.

-        Line 265 fig 3 – please mention in your figure or text, the other sources of OxS (different from diet) such as exogenous sources (cigarette smoking, radiations, drugs, carcinogens) or endogenous sources (activated inflammatory cells, endothelial cells, respiratory process).

-        Line 269 – diabetic retinopathy it chronic is a microvascular complication, not a macrovascular complication.

-        Line 271 – please correct the diabetic retinopathy here to (as a microvascular lesion).  

-        Line 274-276 – please reformulate this sentence, because you are only enumerating different conditions, apparently making no connection between them. And, on the other hand, is redundant to say that postprandial OxS is modulating OxS in DM.

-         Type 2.

-        Line 296 – correct by resistance of insulin receptors in skeletal muscle, because GLUT4 play a receptor role.

Conclusions

Do not mention figures in your Conclusions nor references.

Please rewrite Conclusions and mention the most important idea from your review, that are useful for DM type 2 mechanisms and that are also useful for treatment perspectives.

Please complete this manuscript with the role of inflammation that is strong interconnected with oxidative stress. Please mention the role of glutathione in prostaglandins and leukotrienes synthesis because these molecules are important in inflammatory process.

I would suggest that you should describe the role of NO and nitrosative stress in type 2 DM pathogenesis, because the NO metabolism is one of the most important mechanism for microvascular and macrovascular complications.

Reviewer 2 Report

Comments and Suggestions for Authors

Well-written paper on the role of OxS in T2DM, as well as the effects of supplements on overall metabolic control in this population.

(1.1) The authors refer to prediabetes as a significant health concern. This can be referenced by mentioning clinical studies linking the prediabetic/obese state on risk for CV diseases.

(1.2) Comment on the inflammatory response in T2DM as a contributing factor to the etiology.

(Figures) Figures 2 and 3 can be expanded for better viewing. For figure 3, both font and arrows can be improved.

A summary table of the effects of supplements on the diabetic state is suggested. This table can include age of T2DM pts, duration of disease, duration of treatment, sample size, and major outcomes.

(Conclusion) Not necessary to include reference to figures. This can be omitted. 

Comments on the Quality of English Language

English is fine. Check for missing commas and periods.

Round 2

Reviewer 2 Report

Comments and Suggestions for Authors

The authors have addressed my comments. I have no other comments. Good work.